# Bifacial Dye-Sensitized Solar Cells Utilizing Visible and NIR Dyes: Implications of Dye Adsorption Behaviour

**DOI:** 10.3390/molecules28062784

**Published:** 2023-03-20

**Authors:** Suraya Shaban, Ajendra K. Vats, Shyam S. Pandey

**Affiliations:** Graduate School of Life Science and Systems Engineering, Kyushu Institute of Technology, 2-4, Hibikino, Wakamatsu, Kitakyushu 808-0196, Japan

**Keywords:** dye-sensitized solar cells, bifacial solar cells, dye-bilayer, NIR dye, adsorption behavior, dye cocktail, squaraine dye

## Abstract

Bifacial dye-sensitized solar cells (DSSCs) were fabricated utilizing dye cocktails of two dyes, Z-907 and SQ-140, which have complementary light absorption and photon harvesting in the visible and near-infrared wavelength regions, for panchromatic photon harvesting. The investigation of the rate of dye adsorption and the binding strengths of the dyes on mesoporous TiO_2_ corroborated the finding that the Z-907 dye showed a rate of dye adsorption that was about >15 times slower and a binding that was about 3 times stronger on mesoporous TiO_2_ as compared to SQ-140. Utilizing the dye cocktails Z-907 and SQ-140 from ethanol, the formation of the dye bilayer, which was significantly influenced by the ratio of dyes and adsorption time, was demonstrated. It was demonstrated that the dyes of Z-907 and SQ-140 prepared in 1:9 or 9:1 molar ratios favoured the dye bilayer formation by subtly controlling the adsorption time. In contrast, the 1:1 ratio counterpart was prone to form mixed dye adsorption; the best performance of the BF-DSSCs was shown when a dye cocktail of Z-907 and SQ-140 in a molar 9:1 ratio was used to prepare a photoanode for 1 h of dye adsorption. The BF-DSSCs thus exhibited PCEs of 4.23% and 3.48% upon the front and rear side light illuminations, a cumulated PCE of 7.71%, and a very good BBF of 83%.

## 1. Introduction

Dye-sensitized solar cells (DSSCs) utilizing more than one dye sensitizer, with the capability of light absorption in different optical absorption windows for panchromatic photon harvesting, are one of the plausible approaches to increasing photoconversion efficiency (PCE) [1,2,3]. At the same time, bifacial (BF) DSSCs are expected to collect more photons per day for a given installed space owing to the photon harvesting from both the front and back sides [4]. The recent past has seen the emergence of bifacial device architecture for silicon solar cells to use improved overall PCE in a day [5]. These solar cells solve solar tracking problems efficiently by harvesting photons from both sides, demonstrating improved solar energy harvesting in a day [6]. The performance of these solar cells is assessed by their figures of merit, such as the bifaciality factor (BFF) and cumulative PCE [7]. Silicon-based bifacial solar cells have already achieved PCEs of 17–24% and BFFs ranging from 74% to 89% [8]. Although the installation of bifacial solar cells is growing, problems such as higher module costs and installation conditions have not yet been optimized to achieve widespread market penetration [9]. Among the next-generation solar cells, DSSCs are one of the strong candidates for realizing bifacial solar cells due to the possibility to control the transparency of TiO_2_ by its particle size and its counter electrode (CE) using an ultrathin and transparent catalytic layer and, finally, by the judicious selection of an electrolyte with the least and very weak optical absorption in the visible wavelength region [10]. A bifacial DSSC consisting of a TiO_2_/SiO_2_ anode and a Pt-based CE was demonstrated with a PCE of 6% after irradiation of light from the front or back [11]. A TiO_2_ anode with a front- and back-illuminated DSSC has demonstrated PCEs of 6.54% and 4.26%, respectively, with a BFF of 65% by replacing the costly Pt counter electrode with a low-cost transparent polyaniline as the CE [12,13]. The fabrication of DSSCs using dye sensitizers that harvest visible light was the main focus of earlier research on BF-DSSCs. However, a wide-wavelength photon harvesting from the visible to near-infrared (NIR) region is inevitable for enhancing the PCE of DSSCs. 

Although efforts have been directed towards attaining wide-wavelength photon harvesting by designing a single sensitizer, the performance of these DSSCs was still very poor, in the range of only 3–4% [13,14]. So, multiple dye sensitizers with complementary optical absorption windows are envisioned to overcome the poor efficiency [15]. Using multiple sensitizers in DSSCs often leads to unwanted inter-dye interactions and hampered photon harvesting [16]. Implementing dye bilayers has been proven beneficial in providing an efficient solution to this intriguing issue, leading to synergistic photon harvesting from both dyes by suppressing the inter-dye interactions [17,18,19]. In these reports, although there was an improvement in the PCE by restricting the unavoidable inter-dye interaction, the process was quite complicated and challenging for the large-area commercial application of DSSCs. In this context, a dye cocktail, formed by mixing two or more dye sensitizers, is a common solvent; although it is industrially viable, the prevention of unfavourable inter-dye interaction is challenging. In an exciting report, Ogomi et al. demonstrated that using two dyes with a differential rate of dye adsorption and different binding strengths on the TiO_2_ surface leads to the formation of a dye bilayer from the dye cocktail [20] Unfortunately, the whole study was based on visible light-sensitive dyes only. The speed of adsorption of dye molecules on the TiO_2_ surface and their rate of desorption from the TiO_2_ surface reveal that a combination of two dyes, one with faster adsorption and a weak binding force and the other with slower adsorption and more robust binding, leads to a dye bilayer formation from the dye cocktail [21]. 

In the work being reported here, sensitizing dyes such as Z907/NK-3705 and SQ-140, with complementary light absorption and photon harvesting in the visible and near-infrared (NIR) wavelength region, respectively, were carefully selected. Their adsorption behaviour and binding strength on the TiO_2_ surface were systematically controlled and analysed to attain the dye bilayer from the dye cocktail. Photoanodes with controlled dye adsorption, using two complementary dyes combined with iodine-based redox electrolytes and ultra-thin Pt-coated FTO as transparent counter electrodes, were used to fabricate and characterize the BF-DSSCs. 

## 2. Results and Discussions

### 2.1. Optical Characterization

The electronic absorption spectra of the dyes were recorded in a 10 μM ethanol solution and are shown in Figure 1a. It can be seen from this figure that the NK-3705 dyes absorb light in the 350–450 nm wavelength region, with an absorption maximum (λ_max_) at 426 nm and a molar extinction coefficient (ε) of 1.1 × 10^5^ dm^3^·mole^−1^·cm^−1^. At the same time, the SQ-140 dye exhibited very sharp and intense light absorption, mainly in the far-red (300–700 nm) wavelength region of the solar spectrum, with the λ_max_ and ε of 670 nm and 1.8 × 10^5^ dm^3^·mole^−1^·cm^−1^, respectively. On the other hand, ruthenium complex dye Z-907 in ethanol exhibited a very weak light absorption, mainly in the visible wavelength region encompassing 300 nm to 700 nm, with the ε of 1.7 × 10^4^ dm^3^·mole^−1^·cm^−1^ at the λ_max_ of 520 nm, which is nearly 1/10 as compared to that of SQ-140 at the absorption maximum λ_max_. A higher extinction coefficient of the sensitizer benefits the transparent/semi-transparent DSSCs since sufficient photons can be easily absorbed even with thin layers of mesoporous TiO_2_. In the solid state, the NK-3705 and SQ-140 dyes exhibited a slight red-shifted λ_max_ at 430 nm and 680 nm, respectively, with spectral broadening. This shift in λ_max_ upon adsorption of dyes on TiO_2_ depends on the nature of the dyes, which vary from very small to large, up to 100 nm [22,23]. Upon adsorption onto the TiO_2_ surface, there is a differential spectral shift depending on the nature of the dye; this has been attributed to several phenomena, such as deprotonation of the anchoring group of the dyes (-COOH), p-stacking interactions, complexation with metal ions, and dye aggregation [24,25,26,27]. 

### 2.2. Adsorption Behaviour of Dyes on TiO_2_ Surface

The investigation of the adsorption behaviour on the surface of the mesoporous TiO_2_ not only helps to obtain information about the optimum dye adsorption for better photon harvesting but also provides valuable information about their aggregation, which must be controlled for the optimal DSSC performance. Taking this into consideration, the adsorption behaviour of the sensitizing dyes under investigation was studied by monitoring the spectral absorption changes in the dyes on the mesoporous TiO_2_ as a function of time, which is shown in Figure 2a perusal of the dye adsorption behaviour of NK-3705 revealed that a monomeric dye absorption peak appearing at 430 nm became suppressed with the enhancement of the additional blue-shifted peak appearing at 415 nm as a function of time. This was attributed to the blue-shifted H-aggregate formation on the TiO_2_ surface. At the same time, squaraine dye SQ-140 also exhibited a similar trend of enhanced H-aggregate formation as a function of the increasing dye adsorption time; the blue-shifted vibronic shoulder appearing at 630 nm became more pronounced, and this aggregate peak became almost similar to that of the monomeric dye absorption peak after 60 min of dye adsorption. The planer π-conjugated molecular framework of NK-3705 and SQ-140 could be responsible for the H-aggregate formation, which is very commonly observed in cyanine and squaraine dyes. This is why bulky chenodeoxycholic acid (CDCA) has most commonly been used to prevent dye aggregation and to improve the photovoltaic performance of the DSSCs [28,29]. Contrary to this, Z-907 showed a monotonous increase in the absorbance as a function of dye adsorption time on the mesoporous TiO_2_ surface. There was no shift in the λ_max_ at 520 nm, indicating that there was no dye aggregation, unlike with NK-3705 and SQ-140. This is attributed to the non-planer 3D molecular structure of Z-907 with two long alkyl chains substituted in the phenyl ring, restricting the dye aggregation. This is further supported by the fact that there is no need to add CDCA along with the dye to prevent dye aggregation and enhance PCE-like squaraine dyes. 

The time-dependent absorption spectral changes for dye adsorption as a function of time for different dyes, as shown in Figure 2, reveal that there is a differential rate of the dye adsorption on the mesoporous TiO_2,_ as shown in Figure 3. It is evident from this figure that there is a saturation in the peak absorbance at about 30 min for the NK-3075 and SQ-140 dyes, with the rate of dye adsorption estimated to be 0.018 abs./min and 0.015 abs./min, respectively. On the other hand, Z-907 depicts a very slow dye adsorption, taking more than 5 h to reach saturation for the same thickness of the mesoporous TiO_2_ layer. An estimated dye adsorption rate of 0.003 abs./min demonstrates that Z-907 exhibits dye adsorption at a rate >15 times slower than that of the other two dyes, NK-3075 and SQ-140, as used in the present investigation. 

### 2.3. Anchoring Stability of Dyes on Mesoporous TiO_2_

The anchoring stability of the dye sensitizers on mesoporous TiO_2_ plays a vital role in imparting stability to the DSSCs and in providing the possibility of forming a dye bilayer from a dye cocktail. We have already demonstrated that using a dye cocktail of two dyes sensitizing with differential rates of dye adsorption and binding strengths leads to the formation of a dye bilayer from the dye cocktail with improved PCE by hampering the unfavourable inter-dye interactions [20]. A dye desorption experiment was conducted to evaluate the anchoring stability of the unsymmetrical squaraine dyes used for the present research on mesoporous TiO_2_. To estimate the total dye loading, after complete dye adsorption it was desorbed entirely from the surface using a solvent mixture of acetonitrile, tert-butanol, water, and NaOH. For the dye stability studies, the dye was adsorbed on TiO_2_ substrate with the respective dyes at room temperature; the amount of dye desorbed was estimated spectrophotometrically. The results of the dye desorption from the photoanodes using different dyes are summarized in Table 1. A perusal of Table 1 shows that under identical conditions of dye desorption on TiO_2_, such as thickness, area, and desorption time, the sensitizing dyes exhibit differential binding strengths on the mesoporous TiO_2_, as estimated by the percentage of removal dyes upon putting them into a mixture of NaOH. It is worth mentioning here that SQ-140 exhibited the weakest binding with the highest rate of dye desorption (1.1 nmol/cm^2^/min), followed by NK-3705 (0.85 nmol/cm^2^/min), then Z-907 (0.38 nmol/cm^2^/min). It is suggested that Z-907 exhibited the highest stability on the mesoporous TiO_2_. NK-3705 exhibited the highest dye loading (101.6 nmol/cm^2^), which was significantly higher than that of Z-907 (69.3 nmol/cm^2^) and SQ-140 (66.1 nmol/cm^2^). Based on the dye desorption studies conducted under identical experimental conditions, the relative binding strength on the mesoporous TiO_2_ was found to follow the order Z-907 > NK-3705 > SQ-140.

### 2.4. Photovoltaic Characterization

To study the implications of the dye adsorption behaviour and control for the photovoltaic performance of the BF-DSSCs, the fabrication and characterization of the BF-DSSCs were conducted utilizing individual single dyes as well as two dyes from their respective dye cocktails consisting of a mixture of two dyes taken in different ratios and subjected to the dye adsorption for different time intervals. Efforts were directed towards investigating the implications of the nature of the dyes under consideration for the optical absorption window and the photon harvesting with regard to the performance of the BF-DSSCs fabricated in terms of BFF and cumulative PCEs.

#### 2.4.1. Bifacial DSSC Utilizing Single Dye

Figure 4 depicts the photovoltaic characteristics of the DSSCs using different sensitizing dyes; the summarization of the photovoltaic parameters in terms of short-circuit current density (Jsc), open circuit voltage (Voc), fill factor (%), and photoconversion efficiency (PCE) is shown in Table 2. It can be seen from Figure 4 and Table 2 that amongst the different sensitizers used, the BF-DSSCs fabricated using NK-3705 exhibited not only the least cumulated PCE but also the smallest BFF. The cause of the lowest PCE is a tiny optical absorption window of about 100 nm (350–450 nm), as shown in Figure 1, leading to a very small Jsc. On the other hand, the highly hampered PCE under rear illumination is attributed to the filtering effect by iodine-based redox electrolyte since the triiodide ion of the electrolyte has been reported to absorb the light in a similar optical absorption window [30]. Therefore, the part of the incoming photons incident under rear illumination gets absorbed by the electrolyte, resulting in highly hampered PCE. On the other hand, owing to the relatively higher optical absorption window, mainly invisible for Z-907 (350–700 nm) and dominant in NIR for SQ-140 (380–760 nm), as shown in Figure 1, it could be attributed to the nearly similar cumulated PCE of about 6.5% for the BF-DSSCs fabricated using these two sensitizing dyes. It is worth mentioning here that the BFF for the BF-DSSCs fabricated using SQ-140 was found to be higher (86%) than the BF-DSSCs based on Z-907 (75%). This can also be explained by considering the electrolyte’s enhanced optical filtering effect in the case of Z-907 compared to that of SQ-140. 

#### 2.4.2. Dye Adsorption Behaviour Using Two Sensitizing Dyes

As discussed previously, panchromatic light absorption and photon harvesting in a broad wavelength region are highly desired for enhancing the PCE of DSSCs. At the same time, a bilayer mode of dye adsorption leads to synergistic photon harvesting by both sensitizers due to the suppressed unfavourable inter-dye interactions. To harness this potentiality in the case of utilizing two sensitizing dyes, a dye’s adsorption behaviour from its dye cocktail was investigated systematically and in detail by observing dye adsorption on mesoporous TiO_2_ by taking photographs from the front side (TiO_2_) and rear side (FTO), and the results are summarized in Figure 5. It can be seen from this figure that in the case of using a mixture of two dyes consisting of Z-907 and NK-3705, there is a contrasting colour difference between the red associated with Z-907 and the light green associated with NK-3705 for the photographs taken from the front and rear sides, demonstrating the formation of a dye bilayer with Z-907 at top and NK-3705 at the bottom. This is attributed to the faster rate of dye adsorption and the weaker binding strength of NK-3705 compared to Z-907, where NK-3705 initially gets adsorbed faster throughout the TiO_2_ film but gets replaced by the slow-moving and strongly binding Z-907 with time, leading to the formation of a dye bilayer from the dye cocktail. Considering the non-complimentary optical absorption and enhanced optical filtering in the case of NK-3705 and Z-907, the utilization of these two dye systems to fabricate BF-DSSCs was not suitable.

On the other hand, it can be seen from Figure 1 that Z-907 and SQ-140 dyes exhibit complementary optical absorption, mainly absorbing light in the visible and NIR wavelength region, respectively. This advantage takes into consideration the detailed investigation of their adsorption behaviours on mesoporous TiO_2_; the investigation was conducted by taking their different ratios and subjecting them to dye adsorption for different time intervals, as summarized in Figure 5. It can be seen from this figure that by controlling the ratio of Z-907 and SQ-140 and dye adsorption time, it is possible to control the dye adsorption that forms a mixed layer and a dye bilayer. For example, in the case of the short time of dye adsorption for a period of 1 h, their 9:1 and 1:1 molar ratios depicted different colours to those of the front and back sides. At the same time, a similar observation was made for a longer time of dye adsorption (>4 h) when they were taken utilizing these two dye systems at a 1:9 ratio to fabricate BF-DSSCs. This could be attributed to the fact that in the case of a single dye, SQ-140 exhibits a rate of dye adsorption about three times higher and a binding about four times weaker on mesoporous TiO_2_ as compared to Z-907, as is shown and discussed in Figure 3 and Table 1, respectively. 

It was found that the rate of dye adsorption for single dyes on mesoporous TiO2 was different compared to the cases when the same dyes were subjected to their adsorption from their dye cocktail. Therefore, the adsorption of the dyes Z-907 and SQ-140 from their different dye cocktail solutions as a function of time was also monitored spectrophotometrically, as shown in Figure 6 (top). At the same time, the rate of the dye adsorption for the corresponding single dyes of the dye cocktail was also estimated and compared by plotting the peak absorbances corresponding to Z-907 (520 nm) and SQ-140 (680 nm) as a function of adsorption time; these are also shown in Figure 6 (bottom). In the case of the dye cocktail of Z-907 and SQ-140 taken in a 1:9 molar ratio, it can be seen that there was initially a very fast adsorption of SQ-140, at a rate of 0.031 A/min, reaching saturation in about 90 min. After 150 min, the absorbance corresponding to SQ-140 started decreasing, suggesting the replacement of initially adsorbed SQ-140 by the slow adsorbing (0.001 A/min) and strongly binding Z-907. In the case of a 1:1 molar ratio of dyes in the dye cocktail, the initial adsorption rates for Z-907 and SQ-140 were estimated to be 0.027 A/min and 0.003 A/min, respectively, indicating an enhancement in the adsorption rate of Z-907 that was nearly 3 times faster compared to that observed in the case of 1:9 molar ratio. Contrary to these two cases, the relative rate of dye adsorption in the 9:1 dye cocktail was approximately the same, with the estimated dye adsorption rates of 0.002 A/min and 0.007 A/min for the SQ-140 and Z-907 dyes, respectively. As discussed in Section 2.2 and shown in Figure 4b, when SQ-140 was subjected to dye adsorption it was only after 40 min that the side vibronic peak appearing at 630 nm became pronounced due to enhanced H-aggregate formation. It is interesting to see that such dye aggregation for SQ-140 was highly diminished in the case of each of the dye cocktails, suggesting that the presence of the bulky and 3D Z-907 with long alkyl sides works here not only as a dye co-adsorber but also like a de-aggregating agent for SQ-140, with similar action performed by the large and bulky CDCA molecules, as discussed in Section 2.2.

#### 2.4.3. BF-DSSC Utilizing Dye Cocktails Z-907 and SQ-140 Dyes and 1 h of Dye Adsorption

As discussed in the previous section, it is possible to attain bilayer dye adsorption from dye cocktails by controlling their ratios and dye adsorption times owing to the differential rate of dye adsorption and binding strength on the mesoporous TiO_2_. Taking this into consideration, BF-DSSCs were fabricated using dye cocktails of Z-907 and SQ-140 in different molar ratios and utilizing the corresponding photoanodes for a short period (1 h) of dye adsorption. The photovoltaic characteristics of BF-DSSCs after the front and back light illuminations are shown in Figure 7a, followed by the photovoltaic parameters in Table 3. 

It can be seen from the figure and table that for a short period of dye adsorption an increase in the ratio of Z-907 leads to the cumulative PCE and BFF reaching a maximum of 7.71% and 83%, respectively, for the dye cocktail of 9:1. This is attributed to the fact that in this ratio there are swift rates of dye adsorption for both of the dyes, as shown in Figure 6c, and the presence of the sufficient amount of both of the dyes in the photoanodes, as clearly shown by the spectral absorption features in Figure 7b. At the same time, the differential colour of the photoanodes from the front (reddish) and rear sides (greenish-black), as shown in Figure 5, indicates the presence of a bilayer type of dye adsorption, leading to synergistic photon harvesting from both of the dye sensitizers. In the cases of the 1:9 and 1:1 dye cocktails, the dominance of SQ-140 and the presence of the mixed dye adsorption could be responsible for the lower observed cumulative PCE. 

#### 2.4.4. BF-DSSC Utilizing Dye Cocktails Z-907 and SQ-140 Dyes and 4 h of Dye Adsorption

The results of the photovoltaic performance for the BF-DSSCs fabricated using dye cocktails of Z-907 and SQ-140 in different ratios, but utilizing the photoanodes subjected to a relatively long period of dye adsorption and a period of 4 h, are shown in Figure 8a; the summarization of the photovoltaic parameters is shown in Table 4. At the same time, the absorption spectra of the photoanodes fabricated under this condition of 4 h of dye adsorption and their different dye cocktail solutions are shown in Figure 8b. 

A perusal of Figure 8a and Table 4 confirms that for a relatively long period of the dye adsorption, the trend was reversed, where an increase in the extent of Z-907 in the dye cocktail led to the decrease in both the cumulative PCE and the BFF, which were found to be 6.15% and 83%, respectively. This is attributed to the presence of both the Z-907 and the SQ-140 adsorbed in the bilayer of dye adsorption, as indicated by Figure 6a; after 120 min of adsorption, there was a decrease in the absorbance corresponding to SQ-140. At the same time, under this condition, an increase in the extent of the slow adsorbing and strongly binding Z-907 led to the replacement of the previously adsorbed SQ-140 by Z-907 from the top surface, resulting in a bilayer formation and synergistic photon harvesting by both dyes. On the other hand, in the case of both the 1:1 and the 9:1 dye cocktails of Z-907 and SQ-140, although both dyes were present it was SQ-140 that was rich in the 1:1, while Z-907 was rich in the 9:1 dye cocktails, as indicated by Figure 8b. Therefore, it was not the presence of both dyes but most probably the mixed dye adsorption leading to hampered photon harvesting that resulted in decreased cumulative PCE. Although we have demonstrated the need and the importance of panchromatic photon harvesting using complementarily visible and NIR dyes and their bilayer mode of dye adsorption for controlling the performance of BF-DSSCs, the overall device performance is still low, and further improvement is undoubtedly required. Further enhancement in the performance of the BF-DSSCs utilizing more efficient dyes working optimally with cobalt-based redox electrolytes is envisioned, while preserving the idea of the complementary dye and bilayer mode of dye adsorption. The use of novel dyes with cobalt electrolytes is expected to enhance not only the cumulative PCE but also the BFF, owing to the lower colour-reducing light-filtering effect of the electrolyte itself, especially during rear side light illumination, which leads to enhanced BFF. Research work considering this aspect is currently in progress and will be reported in the future. 

## 3. Materials and Methods

All of the chemicals and solvents were of analytical/spectroscopic grade and used as such without further purification. Fluorine-doped tin oxide (FTO) glass was purchased from Asahi Glass Co., Tokyo, Japan. Solvents such as acetone, ethanol, isopropanol, and acetonitrile were purchased from Wako Pure Chemicals, Tokyo, Japan. Platisol T for preparing transparent counter electrodes and Ti-Nanoxide (D/SP) with a mixture of anatase TiO_2_ nanoparticles of 15–20 nm and 100 nm, were purchased from Solaronix SA, Aubonne, Switzerland. Organic visible dye NK3705 was purchased from Hayashibara Biochemical Laboratories Inc., Okayama, Japan, and ruthenium-based inorganic dye Ruthenizer 520-DN (Z907) was purchased from Solaronix SA. The electronic absorption spectra of the dyes in ethanol solution and the thin films for the dyes adsorbed on TiO_2_ were recorded using an ultraviolet (UV)-visible-NIR spectrophotometer (JASCO model V560).

### 3.1. Sensitizing Dyes

A green-coloured unsymmetrical squaraine dye, SQ-140, was synthesized and characterized as per our earlier publications and employed as an NIR sensitizer in this study for constructing bifacial DSSCs [31]. The Ruthenizer 520-DN red-dye Z-907 is a hydrophobic dye that very efficiently sensitizes semiconductors with a wide-bandgap oxide up to the 700 nm wavelength region. The NK-3705 yellow dye is a dye that harvests light in the lower wavelength up to 450 nm. The molecular structures of the sensitizing dyes are shown in Figure 9.

### 3.2. Dye Adsorption Behaviour and Its Binding Stability on TiO_2_ Surface

The dye adsorption behaviour of the respective dyes was investigated using 14 μm thick TiO_2_ paste (6.25 cm^2^ area) coated on the glass substrate. The TiO_2_-coated glass substrates were then dipped in a 0.2 mM ethanolic solution of the respective dyes. The extent of the dye adsorption as a function of time (until saturation) was monitored spectrophotometrically by measuring the absorbance of the dyes at their absorption maximum (λ_max_). The relative stability of these dyes on the surface of the mesoporous TiO_2_ was investigated by putting the dye-adsorbed TiO_2_ substrate into a solvent mixture of a 40 mM NaOH solution in a t-butanol: acetonitrile: water: ethanol solution taken in an equal ratio at room temperature and by monitoring the extent of the dyes desorbed from the substrate to the solution. The desorbed dyes were quantitated spectro-photometrically using the standard calibration curves of the respective dyes. 

### 3.3. Fabrication of BF-DSSCs

FTO glass with a sheet resistivity of about 23 Ω/sq. was used as a substrate for the working photoanode and counter electrodes. The cut pieces of FTO glass were first cleaned in an ultrasonic bath with detergent and deionized water (DI). This was followed by acetone and isopropanol (IPA) in an ultrasonic bath. They were sonicated for 10 min in each solvent. The cleaned FTO glass substrates were treated with UV-ozone for 10 min to make the substrate surface hydrophilic. Titanium tetrachloride (TiCl_4_) surface treatment was performed on the photoanode as a pre-and post-treatment of the thin film titanium dioxide (TiO_2_) deposited on the FTO glass by screen printing; this was used as a dye adsorption scaffold and electron transport layer. In order to eliminate the solvents and other binding components found in TiO_2_ paste, the films were sintered in a muffle oven at 450 °C for 30 min. In this work, the thickness of the TiO_2_ film at 14 µm was varied by repeating the coating and sintering process and measuring the thickness with a surface profiler. The dye-loaded TiO_2_ photoanode was rinsed with ethanol and air-dried before the DSSCs were made. An ultrathin layer of Pt was coated on the FTO glass to create the transparent counter electrode by spin-coating a Platisol T at 1500 RPM for 30 s, followed by 30 min of sintering in a muffle furnace at 450 °C. The iodine-based redox electrolyte (I^−^/I_3_^−^) was composed of LiI (0.5 M), iodine (0.05 M), t-butyl pyridine (0.5 M), and 1,2-dimethyl-3-propylimidazolium iodide (0.6 M) in acetonitrile; it was sandwiched between the working and the counter electrodes to complete the BF-DSSCs and sealed with UV resin.

### 3.4. Photovoltaic Characterization of the BF-DSSCs

The photovoltaic performance of the fabricated bifacial BF-DSSCs was evaluated using a solar simulator (CEP-2000 Bunko Keiki Co., Ltd., Osaka, Japan) fitted with a xenon lamp (Bunko Keiki BSO-X150LC, Osaka, Japan) employed as a source of simulated solar irradiation at 100 mW/cm^2^, AM 1.5 G. After illuminating the DSSCs with identically intense light from the front and back, the photovoltaic performances were measured simultaneously. The device area of 0.25 cm^2^ was fixed during the measurement using a black metal mask. Figure 10 depicts the schematic representation of the BF-DSSC, where the simulated 1 Sun illumination was irradiated from the front and the rear sides. The bifaciality factor (BFF) of the output of the rear side PCE over the frontal PCE was calculated and presented as a percentage.

## 4. Conclusions

The bifacial DSSC utilization of the two complementary lights absorbing in the visible (Z-907) and NIR wavelength region (SQ-140) for panchromatic photon harvesting was fabricated and subjected to photovoltaic characterization. The investigations of the adsorption rates of the dyes and their binding strengths revealed that the Z-907 dye showed a rate of dye adsorption that was about >15 times slower and a binding that was 2–3 times stronger on mesoporous TiO_2_ compared to the NK-3705 and SQ-140 dyes used for the present investigation. The relatively low cumulated PCE and BBF observed for the BF-DSSCs fabricated using NK-3705 was attributed to a narrow optical absorption window (300 nm–400 nm) along with the strong filtering effect by the iodine-based redox electrolyte during rear side light illumination, owing to their matching optical absorption window. Utilizing the mixture of two dyes, Z-907 and SQ-140, it was demonstrated that it is possible to attain the dye bilayer formation in the dye cocktail solution, which was significantly influenced by the ratio of dyes and adsorption time. The bilayer mode of dye adsorption was proven to be beneficial, leading to synergistic photon harvesting from both dyes by suppressing the inter-dye interactions. Adjusting the adsorption period showed that the dye of Z-907 and SQ-140 produced in a 1:9 or 9:1 molar ratio is favourable for building dye bilayers. In contrast, its 1:1 molar ratio counterpart was prone to form mixed dye adsorption. The best performance of the BF-DSSC was shown when a dye cocktail of Z-907 and SQ-140 in a 9:1 molar ratio was used to prepare the photoanode for 1 hour of dye adsorption. BF-DSSC thus exhibited PCEs of 4.23% and 3.48% upon the front and rear side light illuminations, having cumulated a PCE of 7.71% and a very good BBF of 83%. The utilization of a more efficient visible dye which is compatible with a less-coloured cobalt-based redox electrolyte and its combination of NIR dyes is envisioned to fabricate highly efficient BF-DSSCs with pronounced cumulated PCE and BFF, and current efforts in this direction are in progress.

## Figures and Tables

**Figure 1 molecules-28-02784-f001:**
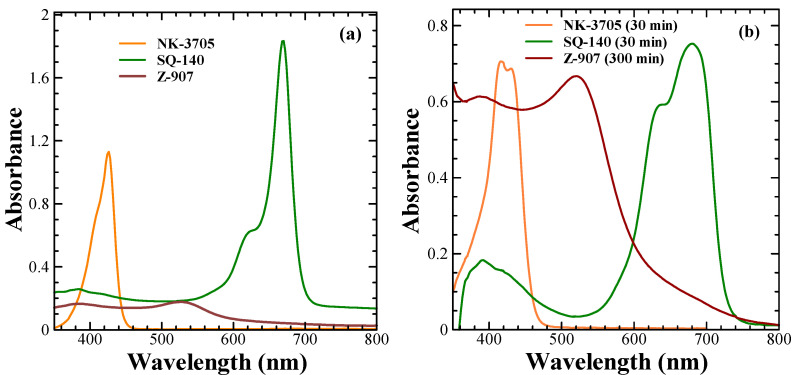
Electronic absorption spectra of NK-3705, Z-907, and SQ-140 in the solution of 10 µm ethanol solution (**a**) and solid-state absorption spectra of these dyes adsorbed onto TiO_2_ thin films (**b**). The values shown in parentheses are the time taken for dye adsorption on 6 µm transparent TiO_2_ thin films to measure the solid-state absorption spectra.

**Figure 2 molecules-28-02784-f002:**
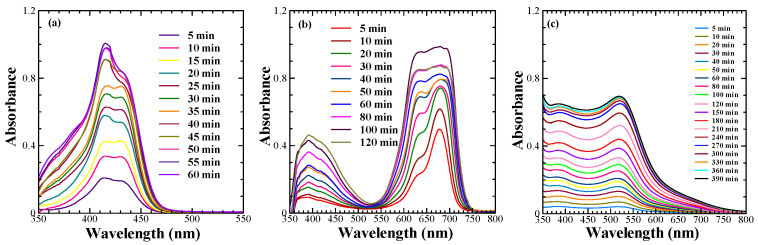
Adsorption behaviour of sensitizing dyes such as NK-3705 (**a**), SQ-140 (**b**), and Z-907 (**c**) on 6 µm thick transparent TiO_2_.

**Figure 3 molecules-28-02784-f003:**
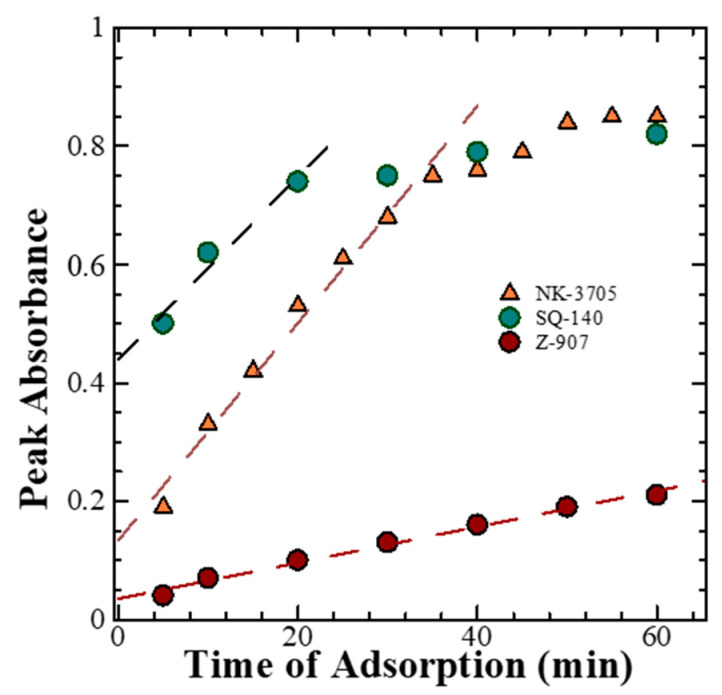
Rate of dye adsorption for NK-3705, Z-907, and SQ-140 on TiO_2_ and the comparison of NK-3705, Z907, and SQ140 single-dye rate of adsorption.

**Figure 4 molecules-28-02784-f004:**
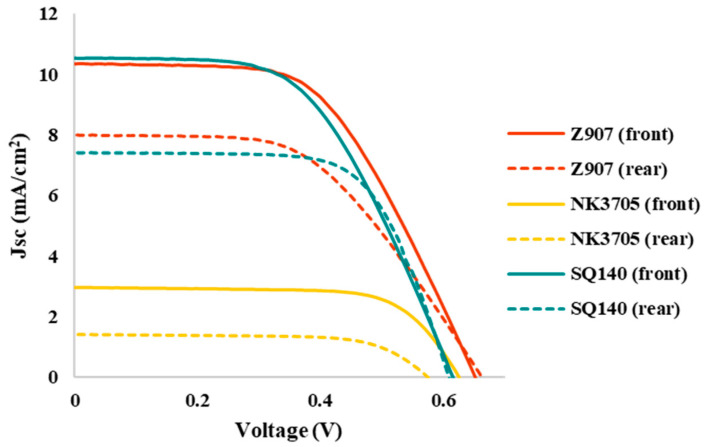
Photovoltaic characteristics of the BF-DSSCs based on TiO_2_ photoanode fabricated utilizing different sensitizing dyes after front-side light irradiation. Corresponding dotted lines represent the J-V characteristics after light illumination from the rear side of the DSSCs.

**Figure 5 molecules-28-02784-f005:**
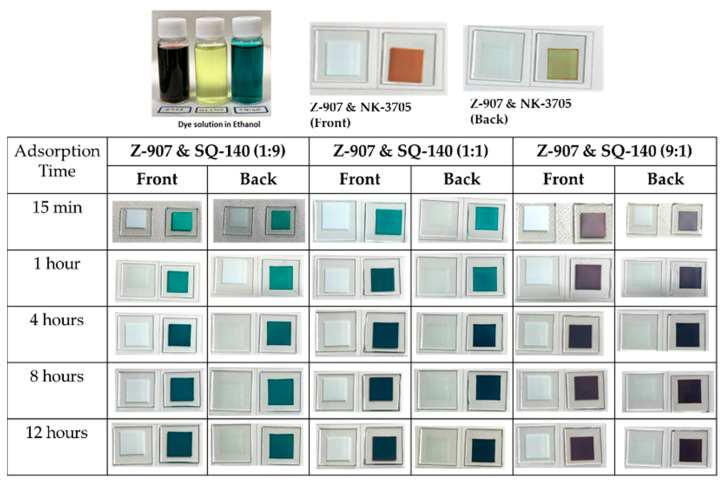
Photographs of the adsorption behaviour of dyes using dye cocktails of two sensitizing dyes with varying molar ratios on the mesoporous TiO_2_.

**Figure 6 molecules-28-02784-f006:**
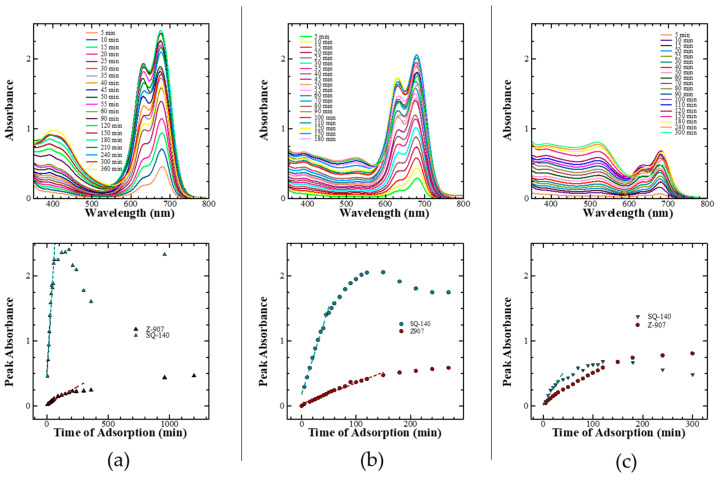
Change in the absorption spectra (top) and peak absorbance corresponding to Z-907 and SQ-140 as a function of time (bottom) for the dye adsorption on mesoporous TiO_2_ from the dye cocktails of two sensitizing dyes taken in different molar ratios (**a**) 1:9, (**b**) 1:1, and (**c**) 9:1.

**Figure 7 molecules-28-02784-f007:**
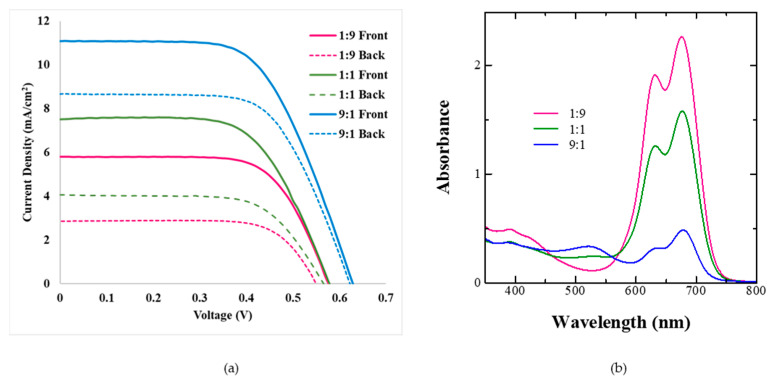
Photovoltaic characteristics of the BF-DSSCs based on the different dye cocktails of Z-907 and SQ-140 in different molar ratios with dye adsorption for a short period (1 h) after front and rear simulated 1 Sun irradiation (**a**) and corresponding absorption for photoanodes taken from different cocktails after 1 h of dye adsorption (**b**).

**Figure 8 molecules-28-02784-f008:**
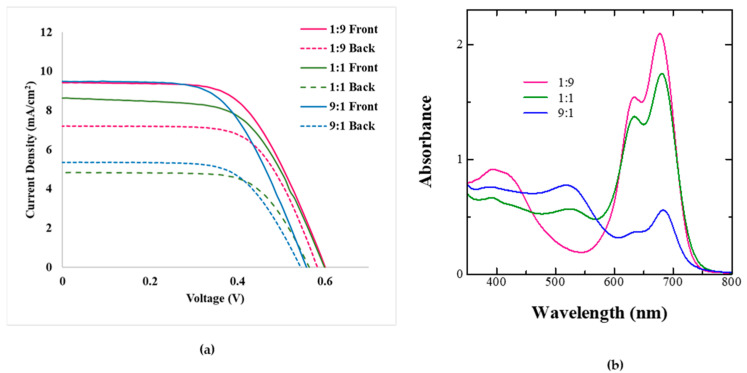
P Photovoltaic characteristics of the BF-DSSCs based on the different dye cocktails of Z-907 and SQ-140 with dye adsorption for 4 h after the front and rear simulated 1 Sun irradiation (**a**) and corresponding absorption for photoanodes taken from different cocktails after 4 h of dye adsorption (**b**).

**Figure 9 molecules-28-02784-f009:**
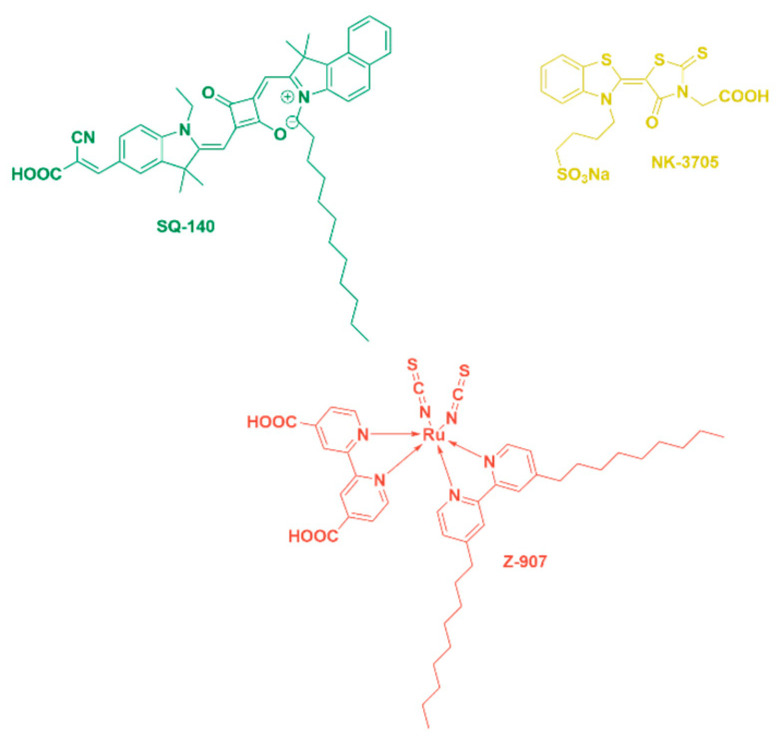
Chemical structures of visible and NIR dyes utilized for the fabrication of BF-DSSCs.

**Figure 10 molecules-28-02784-f010:**
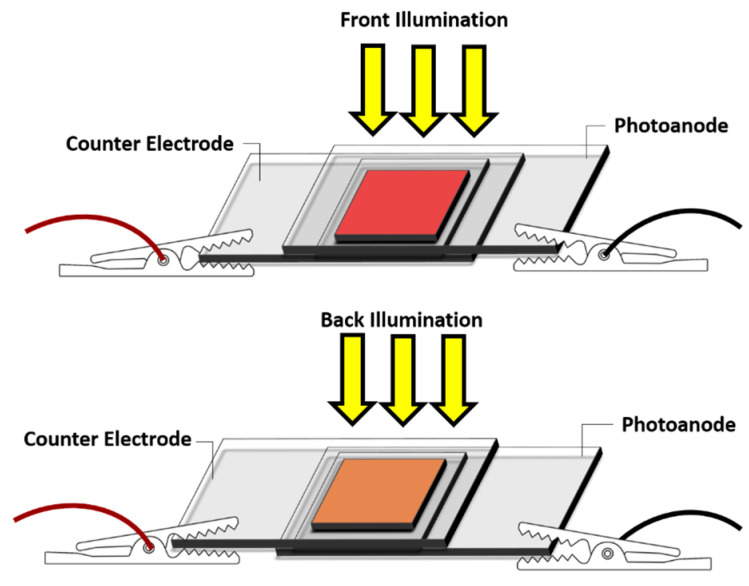
Schematic representation of the structure of BF-DSSC and photovoltaic performance evaluation after simulated solar irradiation.

**Table 1 molecules-28-02784-t001:** Dye desorption behaviour on TiO_2_-coated FTO glass substrate with an area of 6.25 cm^2^ and thickness of 14 µm.

Dye	Total Dye Loading (nmol/cm^2^)	Total Time for Complete Desorption (min)	Rate of Dye Desorption (nmol/cm^2^/min)
NK-3705	101.6	120 min	0.85
SQ-140	66.1	60 min	1.10
Z-907	69.3	180 min	0.38

**Table 2 molecules-28-02784-t002:** Photovoltaic parameters of bifacial DSSCs using single-dye sensitizers after 1 Sun illumination from the front and rear sides of the BF-DSSCs.

Sensitizer	Illumination Side	Voc (V)	Jsc (mA/cm^2^)	FF	Efficiency (%)	Cumulative Efficiency (%)	BFF (%)
NK-3705	Front	0.63	2.99	0.69	1.30	1.86	43
	Back	0.58	1.42	0.68	0.56		
Z-907	Front	0.66	10.37	0.54	3.71	6.50	75
	Back	0.67	8.02	0.52	2.79		
SQ-140	Front	0.62	10.54	0.54	3.53	6.56	86
	Back	0.62	7.44	0.66	3.03		

**Table 3 molecules-28-02784-t003:** Photovoltaic parameters of BF-DSSCs using visible (Z-907) and NIR (SQ-140) dyes for a shorter time (1 h) of dye adsorption, taken in different molar ratios from their dye cocktails.

Dye Cocktail (Z-907: SQ-140)	Illumination Side	Voc (V)	Jsc (mA/cm^2^)	FF	Efficiency (%)	Cumulative Efficiency (%)	BFF (%)
1:9	Front	0.58	5.80	0.67	2.26	3.39	50
	Back	0.56	2.87	0.71	1.13		
1:1	Front	0.58	7.52	0.63	2.74	4.26	56
	Back	0.57	4.06	0.66	1.52		
9:1	Front	0.63	11.09	0.60	4.23	7.71	83
	Back	0.63	8.68	0.64	3.48		

**Table 4 molecules-28-02784-t004:** Photovoltaic parameters of bifacial DSSCs of visible (Z-907) and NIR (SQ-140) dyes for the 4 h of dye adsorption from the different dye cocktail solutions taken in different molar ratios.

Dye Cocktail (Z-907: SQ-140)	Illumination Side	Voc (V)	Jsc (mA/cm^2^)	FF	Efficiency (%)	Cumulative Efficiency (%)	BFF (%)
1:9	Front	0.61	9.43	0.59	3.40	6.15	81
	Back	0.59	7.20	0.65	2.75		
1:1	Front	0.60	8.63	0.60	3.10	4.94	59
	Back	0.57	4.83	0.67	1.84		
9:1	Front	0.56	9.48	0.58	3.09	4.96	61
	Back	0.63	8.68	0.64	3.48		

## Data Availability

Data can be made available upon request.

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
