# Peer review of "Bifacial Dye-Sensitized Solar Cells Utilizing Visible and NIR Dyes: Implications of Dye Adsorption Behaviour"

_molecules, 2023, doi:10.3390/molecules28062784_

Round 1

Reviewer 1 Report

The authors describe in this work bifacial dye-sensitized solar cells employing the mixture of two different dyes (Z-907 and SQ-140), describing a deep and complete characterization. I consider the work interesting and appropriate for publication in Molecules. However, before publication some aspect should be improved:

1.      English mistakes. Some words are bad-written. For example: In line 226 “adsorrption”, in line 246 “fopund”, in line 354 “vibrnic” or in Figure 8 “bsorption”. Please, check them carefully.

2.      In Figure 1, the chemical structures of Z-907 and SQ-140 should be improved. The quality of the image is very low, and different from another dye NK3705 that it is much better.

3.      In Figure 2, the quality of the text is really low. Please, it is necessary to improve it before publishing. The text should be clear.

4.      I don’t totally agree with the sentence “This red-shift in lmax is attributed to interaction of dye molecules with the TiO2 surface” talking about Figure 3. In general, we can always observe a red-shift in solid state in comparison with solution. Could you explain better this statement?

5.      In Figure 3, why Z-907 spectra changes a lot between solution and solid state? The absorbance increases a lot in solid state, however for the other two dyes decrease.

6.      In Figure 5, why Z-907 needs a time so high for saturation in comparison with the other two days?

7.      If I understand fine, in 1 hour adsorption the mixture Z-907/SQ-140 9:1 is better because Cumulative efficiency and BFF are the highest, but in 4 hours is 1:9 the best one, right? Why? Then, what are the best condition for these BF-DSSCs? Please, explain a little better these aspects in the revised version.

8. Finally, what are the advantages of these bifacial solar cells in comparison with the one facial or the conventional architectures in solar cells?

Author Response

Reply to comments of the Referee

(Manuscript ID: molecules-2267906)

Reviewer 1

General Comments

The authors describe in this work bifacial dye-sensitized solar cells employing the mixture of two different dyes (Z-907 and SQ-140), describing a deep and complete characterization. I consider the work interesting and appropriate for publication in Molecules. However, before publication some aspects should be improved:

Reply to the comment: Thank you very much the for your time to review this work and valuable comments. We are ready to revise our manuscript taking all of the suggestions by the reviewer into consideration.

Comment 1. English mistakes. Some words are bad-written. For example: In line 226 “adsorrption”, in line 246 “fopund”, in line 354 “vibrnic” or in Figure 8 “bsorption”. Please, check them carefully.

Reply to the comment: Thanks for the suggestions. We are extremely sorry for the errors and are grateful to the reviewer for pointing them out. In the revised manuscript, such issues have been rigorously examined for any additional existing grammatical and typographical errors, which have been suitably rectified.

Comment 2. In Figure 1, the chemical structures of Z-907 and SQ-140 should be improved. The quality of the image is very low, and different from another dye NK3705 that it is much better.

Reply to the comment: The authors would like to thank the reviewer for the suggestion to improve the quality of the image of the dye structures. Considering the suggestion of the referee, Figure 1 containing the structure of dyes Z-907, SQ-140 and NK-3705 has been remade and included in the revised manuscript.

Comment 3. In Figure 2, the quality of the text is really low. Please, it is necessary to improve it before publishing. The text should be clear.

Reply to the comment: Agreeing with the suggestion of the reviewer to improve the quality of the text in Figure 2, the new schematic has been replaced with better quality in the revised manuscript.  

Comment 4. I don’t totally agree with the sentence “This red-shift in lmax is attributed to interaction of dye molecules with the TiO2 surface” talking about Figure 3. In general, we can always observe a red-shift in solid state in comparison with solution. Could you explain better this statement?

Reply to the comment: Thank you very much for this comment. Agreeing with the suggestion of the referee, it has been explained in more detail along with the inclusion of new references to support this. Such discussion has been newly added as lines 164 to lines 168 on page 4 and 5 of the revised manuscript.

Comment 5. In Figure 3, why Z-907 spectra changes a lot between solution and solid state? The absorbance increases a lot in solid state, however for the other two dyes decrease.

Reply to the comment: Thanks for the comment. Absorption spectra in solution, especially the intensity of the main peak are controlled by the molar extinction coefficient of dyes. Since the same concentration (10 mm) of the dyes is taken for solution spectra, 1/10 of the extinction coefficient of Z-907, as compared to SQ-140 seems very small. This has been already discussed in the manuscript in lines 150-160 on page 4. On the other hand, in the solid state, there is a sufficient number of dye molecules absorbing photons from the incident beam across the film thickness, which is attributed to the differential absorption spectral features in the solution and solid state.   

Comment 6. In Figure 5, why Z-907 needs a time so high for saturation in comparison with the other two days?

Reply to the comment: We agree with this concern of the referee. Actually, the adsorption of dye molecules on the TiO2 surface is controlled by their rate of diffusion in the nanopores, their nature that is overall surface charge, size, and interaction with TiO2. Z-907 being very bulky molecules with long alkyl chains moves very slowly needing a very long time for saturation. Another possibility might the positive charge on Ru central metal of the Z-907 makes attractive interaction with the negatively charged TiO2 surface making its diffusion in the nanopores slower. At the same time, the nature of dye molecules adsorbed on the TiO2 surface also controls the motion of redox ionic species through the TiO2 nanopores. In fact, such work on the differential rate of dye adsorption depending on the structure and nature of the dye molecules has already been published by our group. Please refer to Ogomi, Y. et al, Thin Solid Films, 519, 1087-1092 (2010) and Kawano et al, RSC Advances, 5, 83725 (2015).

Comment 7. If I understand fine, in 1 hour adsorption the mixture Z-907/SQ-140 9:1 is better because Cumulative efficiency and BFF are the highest, but in 4 hours is 1:9 the best one, right? Why? Then, what are the best condition for these BF-DSSCs? Please, explain a little better these aspects in the revised version.

Reply to the comment: Thank you very much for this comment. It has already been explained in the manuscript that there is a need for wide wavelength photon harvesting, which is being carried out by the use of two dyes Z-907 and SQ-140 in this present work. During dye adsorption, the presence of both dyes in optimal amounts is also required. At the same time, using a dye cocktail and having synergistic photon harvesting by both of dyes their adsorption in bilayer fashion (1st dye followed by 2nd dye) to control un-avoidable inter-dye interactions. Since the rate of diffusion and the binding strength of Z-907 and SQ-140 is quite different as discussed in section 3.3.2 of the manuscript, we can control their amount and mode of adsorption (mixed versus bilayer) on the TiO2 surface by controlling their ratio and the time for the dye adsorption. Such optimal conditions like sufficient presence of both of dyes and their bilayer mode of dye adsorption in the dye cocktail of Z-907/SQ-140 met in 1 hour for a 9:1 ratio, while it takes 4 hours for the 1:9 ratio leading to enhancement of the device performance of the BF-DSSCs.    

Comment 8. Finally, what are the advantages of these bifacial solar cells in comparison with the one facial or the conventional architectures in solar cells?

Reply to the comment: We agree with this concern of the referee. The authors would like to enlighten the advantages of bifacial solar cells in comparison with conventional mono-facial solar cells. The current trend, the silicon solar cell is moving from mono-facial to bifacial and it has been demonstrated that the bifacial silicon solar cell is better in overall power harvesting in a day. At the same time, bifacial solar cells not only avoid solar tracking but also space saving.  Moreover, it can also use diffused and reflected light also simultaneously.

 Forecast of the worldwide market shared for bifacial solar cell technology according to the International Technology Roadmap for Photovoltaic (ITRPV) - 11th Ed., April 2020.

Finally, we are grateful to the referee for his valuable comments and suggestions, which enabled us to improve the quality of this manuscript. The necessary changes made, either pointed out by the referees or corrections of English, are highlighted in yellow colour for the perusal of the referee as well as the editor.

Reviewer 2 Report

Please refer attachment.

Author Response

Reply to comments of the Referee

(Manuscript ID: molecules-2267906)

Reviewer 2

General Comments

The article presents the fabrication of bifacial dye-sensitized solar cells (DSSCs) utilizing dye cocktails of two dyes Z-907 and SQ-140 having complementary light absorption and photon harvesting in the visible and near-infrared wavelength regions for panchromatic photon harvesting. However, some recommendations should be taken into account for publication:

  • The overall presentation quality is acceptable. The article is, on the whole, well-written; the English used in the paper is understandable.
  • The following are some of the article's strengths:
  1. The research looks into the use of bifacial dye-sensitized solar cells (DSSCs) for

panchromatic photon harvesting using complementary dyes in the visible and near-infrared wavelength regions, which could lead to improved efficiency.

  1. The study of dye adsorption rates and binding strengths on mesoporous TiO2 provides

important insights for the development of more efficient and stable DSSCs.

  1. The formation of a dye bilayer and identification of the optimal ratio of the two dyes for

bilayer formation could aid in the development of more efficient BF-DSSCs.

Reply to the comment: Thank you for your generosity in taking the time to read this work and for your insightful remarks and for appreciating the work. We are ready to revise our manuscript, considering all of the reviewer's recommendations.

  • Some concerns must be addressed as follows:
  1. The relatively low PCE of the BF-DSSCs fabricated in this study using the dyes suggests that further optimization is required to achieve high efficiency.

Reply to the comment: The authors completely agree with this concern of the referee that further enhancement in the performance of the BF-DSSCs utilizing more efficient dyes working optimally with cobalt-based redox electrolyte is required while preserving the idea of complementary dye and bilayer mode of dye adsorption, which the major emphasis of the work. The Use of novel dyes with cobalt electrolytes is not only expected to enhance the cumulative PCE but also the BFF owing to its less colour-reducing light filtering effect by the electrolyte itself, especially during rear side light illumination leading to the enhanced BFF. Research work considering this aspect is currently in progress and will be reported in future. Such discussion has been newly added as lines 422-432 on page 13 of the revised manuscript.

  1. The research only looked at two dyes, which may not be the most efficient or cost-effective option for large-scale production.

Reply to the comment: The authors agree with the reviewer that this might not be the most efficient or cost-effective option for a large-scale production but to understand the issues related to the BF-DSSCs like how to enhance the PCE and BFF using model dyes absorbing photons from visible to NIR wavelength region. Development and utilization and utilization more efficient dyes along with their functioning with less coloured cobalt-based redox electrolytes are certainly expected to enhance the cumulated PCE as well as BFF, keeping the idea of complementary light absorption in wide wavelength region and attaining the synergistic photon harvesting by both of dyes adsorbing them in the bifacial mode of dye adsorption as demonstrated in this work.  

  1. The study did not investigate the fabricated BF-DSSCs' long-term stability, which is an important factor to consider for practical applications.

Reply to the comment: We completely agree with this concern of the referee that the long-term stability of the device must be ascertained for practical applications and industries and academic institutions are working in this direction too. As it is well known that the most efficient DSSCs reported so far use liquid electrolytes, preventing solvent evaporation is challenging for long-term stability. In this context, companies like Fujikura in Japan have developed two-stage double glass sealing technology to prevent electrolyte leakage and demonstrated the outdoor stability of DSSC panels for >3.5 years and efforts to enhance more are under progress. At the same time, the development of solid-state DSSCs with enhanced efficiency is also in progress. Certainly, long-term stability must be addressed but the theme of the present work is to address enhancing the PCE and BFF issues of the BF-DSSCs as the 1st step.

Overall Merit:

The article certainly has some merit. For the rest, I believe that the article is organized in a logical and understandable manner.

Finally, we are grateful to the reviewer for his valuable comments and suggestions, which enabled us to improve the quality of this manuscript. The necessary changes made, either pointed out by the referee or corrections of English, are highlighted in yellow for the perusal of the referee as well as the editor.